# Lossless Medical Image Compression by Using Difference Transform

**DOI:** 10.3390/e24070951

**Published:** 2022-07-08

**Authors:** Rafael Rojas-Hernández, Juan Luis Díaz-de-León-Santiago, Grettel Barceló-Alonso, Jorge Bautista-López, Valentin Trujillo-Mora, Julio César Salgado-Ramírez

**Affiliations:** 1Ingeniería en Computación, Universidad Autónoma del Estado de México, Zumpango 55600, Mexico; rrojashe@uaemex.mx (R.R.-H.); jbautistal@uaemex.mx (J.B.-L.); vtrujillom@uaemex.mx (V.T.-M.); 2Centro de Investigación en Computación, Instituto Politécnico Nacional (CIC-IPN), Mexico City 07700, Mexico; 3Escuela de Ingeniería y Ciencias, Tecnológico de Monterrey, Pachuca 42083, Mexico; gbarcelo@tec.mx; 4Ingeniería Biomédica, Universidad Politécnica de Pachuca(UPP), Zempoala 43830, Mexico

**Keywords:** lossless image compression, image transform, difference transform

## Abstract

This paper introduces a new method of compressing digital images by using the Difference Transform applied in medical imaging. The Difference Transform algorithm performs the decorrelation process of image data, and in this way improves the encoding process, achieving a file with a smaller size than the original. The proposed method proves to be competitive and in many cases better than the standards used for medical images such as TIFF or PNG. In addition, the Difference Transform can replace other transforms like Cosine or Wavelet.

## 1. Introduction

The use of images has been beneficial for human beings in every aspect of their lives. For every individual, it is very important to have visible evidence of their environment because it provides information that will allow them to make informed decisions. For instance, monitoring medical images is crucial because it equips healthcare professionals with information to assist their patients, with the aim of improving their quality of life. For this reason, technology specialists and healthcare professionals have developed computer-aided systems that rely on image processing [1,2,3,4,5,6] to give better diagnoses [7,8]. Many machine learning and deep learning researchers base their decisions on image databases and other types of databases that could reveal disease information with the goal of handing tools to healthcare professionals so that they come to better conclusions [9,10,11,12,13,14,15,16].

Computer-aided medical systems generate images with higher resolution and a better bit depth; thus, the information which must be processed is higher, especially when 3D scanning technology is used [17,18,19]. Medical imaging also has a defined graphic format called digital imaging and communication in medicine (DICOM) [20].

As is evident, images within the medical area are vital. However, their use is very sensitive. There are mainly two important concerns in the use of medical imaging. The first concern is that images take up a lot of space on devices and consume a lot of time when transmitted by media such as the Internet, so it is necessary to compress them; in doing so, there is a risk of losing important information and in the medical arena, to lose this type of information is restricted by law [15,16]. To address this issue, researchers have developed methods of lossless image compression to be used in medical pursuits and other areas [17,18,20,21,22,23,24,25,26,27,28,29,30,31,32,33]. The second issue is how to eliminate acquisition of noise in images, a topic that has been the inspiration for much research [34,35,36,37,38,39].

Data compression is a mechanism that removes or encodes information with the objectives of reducing storage space and increasing the transmission speed in communication networks [40]. Image compression can be lossy and lossless. Lossy compression removes information to reduce storage space and when reconstructing the information the result approaches the original data. Lossless compression encodes data with a certain amount of information, reducing storage space and, by decoding, reconstructs the original data [22,41]. Lossless compression is the goal of many researchers [16,17,18,20,21,22,23,24,25,26,27,28,29,30,31,32,33].

State-of-the-art lossless compression methods have been published, and two in particular stand out—Wavelets [17,20,24,27,28,32,33] and deep learning and machine learning methods have obtained very promising results [42,43,44,45,46,47,48,49,50,51].

The aim of this paper is to present a new lossless medical image compression algorithm. The Difference Transform algorithm is designed in such a way that, if there is a lot of information, its compression will be greater. Therefore, if the images contain a lot of information, such as RGB images, the compression will be greater than commercial formats such as JPEG, PNG, and TIFF. Another advantage of the algorithm is its implementation because it is simple; this will be shown later in the paper. The algorithm shows disadvantages with 8-bit grayscale images. The problem to solve, in this paper, is to find a new model of lossless compression that overcomes the TIFF and JPG graphic formats that are widely used in medical image compression. The proposed algorithm, as will be shown in the results section, is an algorithm that can be taken into account for lossless medical image compression. To summarize, we will present in this paper a new state-of-the-art method based on the transformation of differences for the lossless compression of medical images.

### Related Works

In this section, we will describe works related to lossless image compression that provide relevant information for the Difference Transform algorithm. For the purposes of this paper, we will classify the application of lossless image compression into two classes; that of natural or conventional compression images and that of compression medical images. The aim of this classification is to highlight that existing state-of-the-art methods or algorithms apply to any image regardless of what it represents. The importance of emphasizing medical images is because in these there is a very important impact and meaning for the human being. In addition, we emphasize that the use of existing lossless compression algorithms, as well as the algorithm we propose are very useful for the storage and transmission of medical images due to the volume of information they contain and the importance of the losslessness of information.

A method of compression in non-medical images is presented by Báscones and et.al. They show a new lossless compression algorithm applied to hyperspectral images based on the Wavelets transform. The algorithm spectrally decorrelates the image by vector quantification, carries out the analysis of main components, and applies the JPEG2000 algorithm to the main components, taking advantage of the fact that the dimensionality reduction preserves more information. The algorithm gets a 1- to 3-dB increase in signal-to-noise ratio for the same compression ratio just by using its PCA + JPEG2000 algorithm, while raising compression and decompression by more than 10% [21].

Further relevant work in the compression of non-medical images is the one shown in [52].They propose an alternative but efficient coding algorithm that uses Huffman’s coding algorithm. The proposed algorithm reduces the number of bits that are symbolized by long bitcode words by using Huffman’s encoding algorithm. The algorithm validates it with three different groups of images. The algorithm successfully encodes image compression operations. Depending on the image characteristics, the algorithm achieves a 2.48% to 36% compression. Another interesting compression algorithm is the one proposed by Starosolski where a new transformation based on the discrete transformed Wavelet is presented. This transformation is built adaptively to the image by using heuristics and entropy estimation. Compared to unmodified JPEG2000, it improved the compression ratios of photographic and non-photographic images, on average, by 1.2% and 30.9%, respectively [24].

The algorithms mentioned above were based on the use of transform and Huffman’s coding method. Now we will cite an interesting work of convolutional neural networks for the losses of non-medical image compression. In [23], a low-complexity compression approach to multispectral imaging based on convolution neural networks is proposed (CNNs). They create a new spectral transformation by using CNN. Their experimental results show that the proposed method improved compression efficiency by 49.66%.

The latter work presents an interesting submission on CNN for losses in non-medical compression. Nonetheless, the works cited above show two distinct areas that treat image compression, implementing transformations and deep learning. It is interesting to see that these two areas are generating important results in lossless images compression, and it will be seen in the following works that this type of algorithm is applicable to medical images with notorious results.

Reference [20] shows a method using second-generation Wavelets and the set partitioning in hierarchical trees (SPIHT) algorithm. The experiments shown from 3D DWT tomographic images indicate that the bit width of the wavelet filter coefficients could be significantly reduced to obtain high-quality medical images. The algorithm shows that at low bit rates, its algorithm called bandelet-SPIHT yields significantly better results compared to some coding techniques, such as the H.26x family (i.e., H.264 and H.265), ensuring that this is appropriate for medical use.

Another lossless medical image compression algorithm is the one presented in [28]. This work presents a hybrid method that enhances JP3D compression of volumetric medical images. The method is based on the discrete wavelet transformation (DWT). It applies reversible noise removal and elevation steps with a three-dimensional (3D) DWT step jump and builds a hybrid transformation that combines 3D-DWT with prediction. The authors propose practical compression schemes that improve the compression ratio by up to 6.5%.

Now then, in [53] it shows a method that uses combinations of algorithms that compress X-ray images, which are registered in the state of the art. The results show that the right combination of used compression algorithms submit high percentages of lossy and lossless compression. The algorithms that obtained the best compression were the RLE-Compressed, Discrete Cosine Transform (DCT), and the Discrete Wavelets Transform (DWT). Their results show that under the criteria of peak signal-to-noise ratio, the DCT obtained 89.98 and the DWT obtained 54.77, highlighting that these two algorithms are the ones that had the best performance.

In [54] a non-iterative method of lossless dental imagery compression is proposed, based on the discrete cosine transformation and the optimization of the partition scheme, procuring improvements in the compression of the images used. They achieved compression ratio values between 7.5 and 20.6, depending on the image format with which they were compared, one of those being JPEG2000. In [55], it proposes a method of compression of endoscopic images that is based on the 3D discrete cosine transformation and proposes an adaptive filter of frequency domain that is fundamental for compression. The results show that the proposed method reaches a compression ratio of 22.94:1 with the peak signal to noise ratio of 40.73 dB.

As can be seen, the works related to the lossless medical image compression demonstrate that the use of DTC and DWT are cornerstones for compression, in addition to the quintessential JPEG2000 method. This is relevant because we propose an algorithm based on the transform of differences that outpaces JPEG2000 and that the implementation of the method is simpler than that of DTC and DWT.

## 2. Materials and Methods

### 2.1. Laplacian Pyramid

A powerful but conceptually simple structure that can be used for the representation of images in more than one dimension is the Laplacian pyramid or pyramidal multiresolution [56]. These structures were originally developed for applications in computer vision and image compression. An image pyramid is a collection of images with decreasing resolutions arranged in a pyramid shape [57]. As shown in Figure 1, the base of the pyramid contains a high resolution representation of the image to be processed and the peak contains a low resolution approximation. As one moves up the pyramid, the size and the resolution decreases. Taking the *J* base level with a size of $2^{J}\times 2^{J}$ or N×N, and intermediate levels taking a size $2^{j}\times 2^{j}$, with 0≤j≤J.

Given a sequence x(n),n∈Z it is possible to derive a signal in low resolution by a low-pass filtering g(n) and then applying subsampling by two, thus doubling scale analysis. The result is a signal y(n) given by
(1)$$y\left(n\right)=\sum_{k=-\infty }^{\infty }g\left(k\right)x\left(2n-k\right).$$

The change in resolution is obtained by the low-pass filter (loss of high-frequency detail). The scale change is due to the subsampling by two, because a displacement by two in the original signal x(n) results in a displacement by one in y(n). Based on the filtered and subsampled version of x(n), it is possible to find an approximation x(n) of the original. This is done first by oversampling y(n) by two, because it is necessary to have a sign of the same scale as the original for comparison:(2)$$y^{\prime }\left(n\right)=y\left(n\right),\;\;y^{\prime }\left(2n+1\right)=0.$$

Then $y^{\prime }\left(n\right)$ is passed by a filter impulse response $g^{\prime }\left(n\right)$ to obtain the approximation to a(n): (3)$$a\left(n\right)=\sum_{k=-\infty }^{\infty }g^{\prime }\left(k\right)y^{\prime }\left(n-k\right).$$

Of course, generally, a(n) will not be equal to x(n); therefore it is possible calculate the difference between a(n) and x(n) as
(4)d(n)=x(n)−a(n).

As shown, x(n) can be reconstructed by adding a(n) and d(n). However, some redundancy exists, because a signal with sampling frequency $f_{s}$ is expressed in the two signals y(n) and d(n) with sampling frequencies $\frac{f_{s}}{2}$ and $f_{s}$, respectively. The separation of the original signal x(n) into an approximation a(n), plus the sum of a signal containing the detail d(n) is conceptually important, because changing the resolution and some other relationships are part of the multiresolution analysis.

### 2.2. Subband Coding

In subband coding, a signal is a set of elements in limited bands named subbands, which can used to recover the original image without error. First developed for voice compression [57,58], each subband is generated by band-pass filtering of the input; then, the resulting bandwidth is smaller than in the original image. Furthermore, each subband may be subsampled without loss of information.

The process for subband coding is part of the pyramidal multiresolution scheme. The signal obtained from the low-pass filtering is the same, but instead of a difference signal, it calculates a detail aggregate through a high-pass filtering x(n) by using a filter with impulse response h(n), followed by subsampling by two. Intuitively, it is clear that the added detail of the low-pass approximation has to be a high-pass signal. In addition, if g(n) is an ideal halfband filter lowpass, then a half-band filter is ideal for a perfect representation of the original version with the two undersampled versions.

Then x(n) can be recovered from the two filtered and undersampled versions $y_{0}\left(n\right)$ and $y_{1}\left(n\right)$ by g(n) and h(n), respectively. These are necessary for both the oversample as to filtering by $g^{\prime }\left(n\right)$ and $h^{\prime }\left(n\right)$, respectively, and finally adding both, as shown in Figure 2. Conversely, in the pyramidal case, the reconstructed signal $\hat{x}\left(n\right)$ is not equal to x(n), unless the filters have specific characteristics. The simplest case occurs when analyzing the reconstructed signal is identical to the original ($\hat{x}\left(n\right)=x\left(n\right)$). If this happens, then it is said that the filters have the property of perfect reconstruction.

Because the attainment of the perfect reconstruction filter is the subject of much research, it is assumed to have a finite impulse response filter (FIR). So it turns out that the low-pass and high-pass filters are related to
(5)$$h\left(L-1-n\right)={\left(-1\right)}^{n}g\left(n\right),$$
where *L* is the filter length.

Now, the filter bank in Figure 2, which computes convolutions followed by subsampling by two, evaluates the inner product of the sequence x(n) and the sequences g(2k−n), h(2k−n):(6)$$y_{0}=\sum_{n}x\left(n\right)g\left(2k-n\right),$$
(7)$$y_{1}=\sum_{n}x\left(n\right)h\left(2k-n\right),$$
and reconstructing x(n) is given by
(8)$$x\left(n\right)=\sum_{k=-\infty }^{\infty }\left[y_{0}\left(k\right)g^{\prime }\left(2k-n\right)+y_{1}\left(k\right)h^{\prime }\left(k-n\right)\right].$$

## 3. Differences Transform

A major reduction in the amount of data representing an image is obtained by eliminating or decreasing the redundancy between them. The best way to achieve this is by using some cane transformation of the processed image. In this paper, we propose to use a difference transform, observing that the redundancy can be eliminated by correlation of the analyzed image. For this, the analysis is based on the relationship between three adjacent samples. The difference transform operator can be developed within the discrete plane as follows.

Given a sample sequence x(n), it is possible to know the value of any of the samples by means of neighbors. This is achieved by dividing the sequence into two parts. The first part will contain an undersampled-by-two version of the original sequence and the second part consists of the values obtained by subtracting the remaining samples with the neighbors average; this can be expressed by Equations (Equation 12) and (Equation 13):(9)y(k)=x(2k),
(10)$$y\left(k+\frac{N}{2}\right)=\left\lfloor \frac{x(k-1)+x(k+1)}{2}\right\rfloor -x\left(k\right),$$
where *N* is the size of the sequence.

The second part (Equation (Equation 13)) can be considered as follows: (11)$$\begin{array}{ccc} y\left(k+\frac{N}{2}\right) & = & \frac{x(k-1)}{2}+\frac{x(k+1)}{2}-x\left(k\right)\\ & = & \frac{x(k-1)}{2}-x\left(k\right)+\frac{x(k+1)}{2}\\ & = & \frac{1}{2}x\left(k-1\right)+\left(-1\right)x\left(k\right)+\frac{1}{2}x\left(k+1\right)\\ & = & \sum_{l=-1}^{1}h\left(l\right)x\left(k-1\right). \end{array}$$

As shown, the last equality corresponds to the convolution between *x* and *h*; this is considered as the component of a digital filter that performs a filtering on the sequence. Because it is also necessary to make an undersampling by two, it is possible to use Equation (Equation 1), finally, to obtain the Difference Transform, as follows:(12)y(k)=2k,
(13)$$y\left(k+\frac{N}{2}\right)=\sum_{l=-\infty }^{\infty }h\left(l\right)x\left(2k-l\right).$$

The above procedure applies only to the transformation. It is then necessary to have a method to recover the original sequence, i.e., the inverse transformation.

For the inverse transform, we first proceeded with the sequence y(k) consisting of the undersampled original samples concatenated with the differences of the average values, by interleaving them as stated in Equations (Equation 14) and (Equation 15):(14)$$\hat{x}\left(2n\right)=y\left(n\right),$$
(15)$$\hat{x}\left(2n+1\right)=y\left(n+\frac{N}{2}\right).$$

At this stage, the interleaved values $\hat{x}\left(2n+1\right)$ do not correspond to the one original sequence, but they are related with difference of the average values of its neighbors, as follows in Equations (Equation 13) and (Equation 16):(16)$$\hat{x}\left(k\right)=\left\lfloor \frac{\hat{x}\left(k-1\right)+\hat{x}\left(k+1\right)}{2}\right\rfloor -\hat{x}\left(k\right),$$
with k=2n+1, and rewriting the Equation (Equation 12) as: (17)$$\hat{x}\left(k\right)=\sum_{l=-\infty }^{\infty }h\left(l\right)\hat{x}\left(k-l\right)\;\;k=2n+1.$$

This way, the Inverse Difference Transform is represented by Equations (Equation 14)–(Equation 16). Notice that the processes and the transform are very simple, with the involved digital filters in both processes being the same. The coding scheme using a block diagram for the Difference Transform in one dimension is shown in Figure 3.

### 3.1. Difference Transform in Two Dimensions

The Difference Transform in two dimensions for encoding, in case of images, is developed from the transformation in one dimension as detailed below.

In a similar way as in the Wavelet transformation process [59,60], the Difference Transform in two dimensions used a single digital filter, which is the same as the one used in the one-dimensional transformation. In the case of wavelets, the filtering is first done in one dimension and then performed again in the other dimension in orden to obtain the subsets of approximation and detail. For this, the wavelet process used four digital filters, two for each dimensions. On the other hand, the two dimensional process in the Differences Transform is performed by similar filtering processes, but using only three filtering processes instead of four. The encoding method using the Difference Transform is performed as follows: Let f(x,y) be the original image to analyze and $h^{H}$, $h^{V}$, $h^{D}$ be the digital filters, whose dimensions and values are identical and are used to filter the original image. In the first filtering process, $h^{H}$ is used to obtain details or variations between neighboring samples horizontally, by $h^{V}$ in the vertical direction and with $h^{D}$ in diagonally. After the Difference Transform, three subpictures with half-width and half-height dimensions are obtained, and in addition the image in its original form, undersampled by two, is obtained. Thus we have four subimages with the same size, whose order can be exemplified by Figure 4.

The filtering process with $h^{H}$ is only on the *x* axis, and the filtering process with $h^{V}$ on the *y* axis. As in one dimension, subimages with horizontal and vertical details are obtained; for the case of the diagonal details, the direction of the filter is in the *x* and *y* axis. Finally it is necessary to undersample the original image to obtain the two dimensional difference transform as follows: (18)$$\begin{array}{ccc} W^{S}\left(m,n\right) & = & f(2m,2n) \end{array}$$(19)$$\begin{array}{ccc} W^{H}\left(m+\left(\frac{M}{2}\right),n\right) & = & \sum_{l=-\infty }^{\infty }h^{H}\left(l\right)f\left(2m-l,2n\right) \end{array}$$(20)$$\begin{array}{ccc} W^{V}\left(m,n+\left(\frac{M}{2}\right)\right) & = & \sum_{l=-\infty }^{\infty }h^{V}\left(l\right)f\left(2m,2n-l\right) \end{array}$$(21)$$\begin{array}{ccc} W^{D}\left(m+\left(\frac{M}{2}\right),n+\left(\frac{M}{2}\right)\right) & = & \sum_{l=-\infty }^{\infty }h^{D}\left(l\right)f\left(2m-l,2n-l\right) \end{array}$$
where *M* is the width and *N* is the height of the image.

Once the four subimages are obtained, they are arranged similarly to the shape of the wavelet transform, as shown in Figure 5.

The decorrelation can be achieved through the filtered process because in the analysis we can come to the following decision: if the pixel is equal to its two neighbors, it can be removed later and recovered with its neighboring values. However, if the said pixel value is very different from its neighbors, this indicates that it is a part of the detail of the image, and therefore it is necessary to keep the value based on the difference between it and its neighbors. The variations between neighboring pixels of the image—horizontally, vertically, and diagonally—are obtained by using $h^{H}$, $h^{V}$ and $h^{D}$ respectively. The filter structure can be observed in Figure 6.

The Difference Transform algorithm is show in Algorithm 1.
**Algorithm 1:** Differences Transform algorithm (TDiferences function).
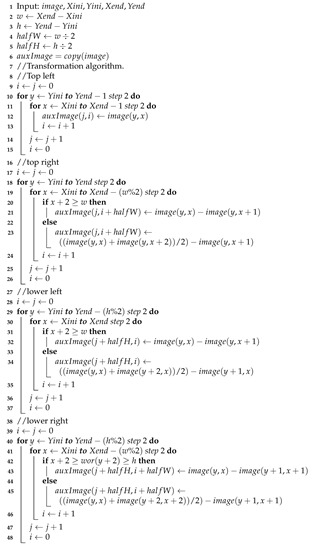


Due to the similarity of the process of wavelet transform with the Difference Transform, it is possible to perform a multiresolution method. However, in a different way from the wavelet process, one of the subimages in the decomposition process is not affected by the filtering process and this is the image that can be affected again by a decomposition process, thereby obtaining a second level of decomposition with its corresponding four subimages, as shown in Figure 7. This procedure can be iterated as many times as required or until we get subimage of 3×3 pixels. The algorithm for multiresolution process is show in Algorithm 2.
**Algorithm 2:** Difference Transform multiresolution algorithm.
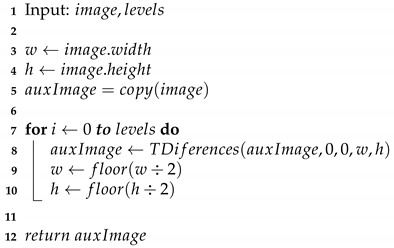


For the recovery of the image, as expected, it is necessary to change the undersampling by the oversampling, also using the filters $h^{H}$, $h^{V}$, $h^{D}$, as in the case of one dimension before performing the filtering process. This process is shown in Figure 8.

The oversampling process inserting specific values for the image before the filtering process is represented by the following equations: (22)$$\begin{array}{ccc} \hat{f}\left(2x,2y\right) & = & W^{S}\left(x,y\right) \end{array}$$(23)$$\begin{array}{ccc} \hat{f}\left(2x+1,2y\right) & = & W^{H}\left(x,y\right) \end{array}$$(24)$$\begin{array}{ccc} \hat{f}\left(2x,2y+1\right) & = & W^{V}\left(x,y\right) \end{array}$$(25)$$\begin{array}{ccc} \hat{f}\left(2x+1,2y+1\right) & = & W^{D}\left(x,y\right). \end{array}$$ Now that we have an original image, it is passed through the three digital filters, whose equations are derived in the same way as the one-dimensional process given in the Equation (Equation 17): (26)$$\begin{array}{ccc} \hat{f}\left(2x+1,2y\right) & = & \sum_{l=-\infty }^{\infty }h^{H}\hat{f}\left(2x-l,2y\right) \end{array}$$(27)$$\begin{array}{ccc} \hat{f}\left(2x,2y+1\right) & = & \sum_{l=-\infty }^{\infty }h^{V}\hat{f}\left(2x,2y-l\right) \end{array}$$(28)$$\begin{array}{ccc} \hat{f}\left(2x+1,2y+1\right) & = & \sum_{l=-\infty }^{\infty }h^{D}\hat{f}\left(2x-l,2y-l\right). \end{array}$$ Thus the inverse Difference Transform is formed by Equations (Equation 22)–(Equation 28). Because in filtering the interleaved sample values depend only on the original samples ($W^{S}$), filtering can be performed in any order. This represents the inverse transformation on one level; consequently, similarly to the decomposition process, it can also be applied to a larger number of levels, as shown in Figure 9, Algorithms 3 and 4.
**Algorithm 3:** Inverse Differences Transform algorithm (TIDiferences function).
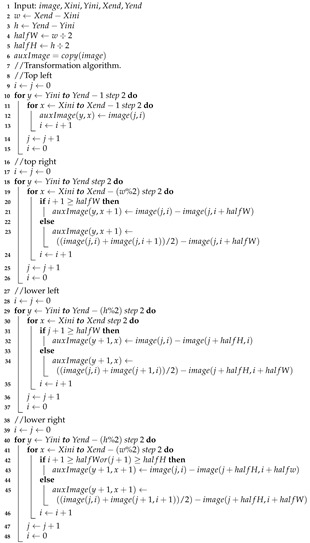


**Algorithm 4:** Inverse Difference Transform multiresolution algorithm.

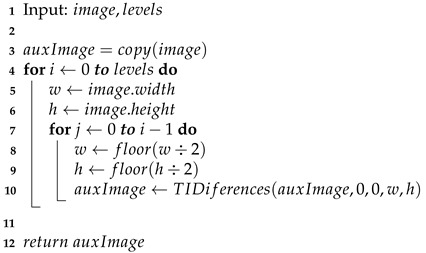



Figure 10 shows the diagram of how the complete coding process is performed. At first, the Difference Transform is applied to the original image and the Huffman encoding is applied to the resulting data, generating the compressed file of the image. In addition, in Figure 4, a numerical example is observed when applying the Difference Transform. The blue color represents the values for $W^{s}$, the orange color represents $W^{H}$, the green color is $W^{D}$, and the yellow color is $W^{V}$. Also, Figure 4 shows the diagram blocks of the decompression process.

## 4. Results

In this section, we will describe the process for the application of 2D TDC in medical images and in some conventional images. The image dataset used for the experimental scheme in this section are comprised of the following:2 classic images (Lena and House) in RGB and grayscale that are referenced in image processing;9 natural images in PGM format of different sizes in 8 and 16 bits in both color and grayscale;6 color images with different sizes and 24 bits that correspond to common examples in image processing; and3 medical imaging datasets. The first dataset contains 612 items corresponding to 24-bit color colonoscopy images captured in original TIFF format. The second dataset contains 850 chest X-ray images, in 24-bit color of the original PNG format. The third dataset contains 517 knee X-ray images (1 and 2 knees) in 24-bit color and original PNG format.

Figure 11 illustrates the process applied to generate the results of this section. As shown in Figure 11, Algorithm 2 is applied to the images, which in turn calls on Algorithm 1, as described in Section 3.1. After applying the Algorithm 2, an array of integer values of the same dimension as the image is generated. These values are between $\left[-2^{N},2^{N}\right]$ where *N* is the number of bits with which the image is represented. To achieve compression in this matrix, it is necessary to apply an encoding method to eliminate data redundancy. In this paper, to check the 2D TDC efficiency, Huffman coding is applied. As a result of applying the encoding method, an images bank with lossless compression is obtained. From this dataset, tables and graphs, which will be shown in this section, are obtained. To verify that the images were losslessly compressed, Algorithm 4 is applied, which in turn invokes Algorithm 3, as described in Section 3.1. Although Figure 11 is described for the medical image dataset, it is also applicable to conventional images.

In order to visually illustrate what happens with the 2D TDC, the following Figures are presented. Figure 12 presents the original images, Figure 13 visually shows the result of applying the Algorithm 2 and finally Figure 14 shows the result of applying the 2D TDC three times.

Once we know what happens with the 2D TDC, we will begin to apply it to a non-medical image dataset in order to demonstrate the compression of the 2D TDC capabilities. After obtaining the results in these images, the algorithm proposed in this paper will be applied to a medical image dataset to show that 2D CDT is a good alternative in images where the information must be kept 100% in light files for its use.

The compression ratio, shown in all Tables, was calculated by the original image file size divided by the compressed image file size (TDC and encoding).

Figure 15 and Figure 16 show a set of 9 images with different sizes, which represent the images in grayscale and in color used, both for 8 and 16 bits. As seen in Table 1 and Table 2, the commercial JPEG-LS algorithm is practically the one with the best lossless compression for this image set. However, the results shown in Table 1 and Table 2 of the 2D TDC are the ones with the least compression. This does not necessarily imply that this algorithm should be discarded because the 2D TDC is designed in such a way that the greater the number of bits used, the greater the compression ratio. The increase in compression is linear; thus, more information means more compression. Due to the nature of the 8-bit images used in this process, the images are already light by definition. The 2D TDC compresses less than the other algorithms, and the impact on the final size is not very significant. This is illustrated in the results shown in Table 3 and Table 4.

Figure 17 shows the difference in compression rates between JPEG-LS and 2D TDC for the 9 images used. It can be highlighted that, with the exception of the first image, which is synthetic, the difference in compression is not remarkable. Figure 17 confirms what was mentioned in the previous paragraph. As there is less information, because they are 8-bit images, the 2D TDC compresses less than the JPEG-LS, but the difference in the compression rate is not significant. Similarly, Figure 18, based on the data in Table 2, shows the same behavior as Figure 17 and the reason is that both sets of images are 8 bit. In regard to image 1, which is the 8-bit synthetic, the colors and grayscales are arranged in such a way that it generates an advantage for the compression algorithms that are different from the 2D TDC, which is why the difference in compression is so marked between the JPEG-LS and the 2D TDC in this image.

Table 3 shows the compression ratio between the algorithms used in this paper. The algorithms are applied to a set of 9 16-bit grayscale images. The 2D TDC obtains the best lossless compression and JPEG-LS is second. It was previously highlighted that Figure 17 and Figure 18 show that the difference between the image compression rates of the JPEG-LS and 2D TDC algorithms was not very noticeable. However, the situation changes, because now that there are more bits, the 2D TDC is compressed more, and the difference in compression rates between the JPEG-LS and the 2D TDC is further apart.

Table 4 shows the compression ratio of a set of 16-bit RGB images. It can be seen that the 2D TDC algorithm is the one with the highest lossless compression. The second highest is the JPEG-LS algorithm (an algorithm widely used in commercial applications). The 2D TDC confirms that the more information the image has, the more it is compressed, and this is graphically shown in Figure 19 and Figure 20. When comparing Figure 17, Figure 18, Figure 19 and Figure 20, it can be seen that the more bits are present drives the difference of the rate compression between the JPEG-LS and 2D TDC to be more significant; that is, the rate compression of 2D TDC is better than the JPEG-LS.

Figure 21 shows a set of 6 24-bit RGB images. The lossless compression algorithms used are PNG, TIFF, and TDC. The TIFF format is added as it is widely used for creating medical imaging datasets. Table 5 and Figure 22 illustrate that TIFF has the best lossless compression on this set of images. The TDC generates the second best compression and lastly is the PNG. The images that make up this set are relatively medium-sized images; however the information content they have is high, which implies the importance of compression. In addition, it is shown that the difference between the compression rate with non-medical images between the TIFF and the TDC is not very big. TIFF seems to be a better compressor, although in medical images, Table 6, Table 7, Table 8 and Table 9 show the opposite.

Concerning medical images, as mentioned above, the most used compression formats are TIFF and PNG, so they are being compared in lossless compression rate with 2D CDT. Figure 23 shows 9 images out of 612 that correspond to extracts from colonoscopy video frames that make up the CVC-ClinicDB dataset. These images are in TIFF format. Table 6 shows some of the selected compression ratio results. In addition to this, Table 6 shows the average, maximum, and minimum of the compression rate. The latter is calculated as follows: the compression of all the images is performed, the minimum compression is obtained, the maximum compression and the compression rates are averaged. The 2D TDC has the best compression.

Table 7 shows the compression ratio for normal (non-COVID-19) 24-bit chest X-ray images with PNG format; taking as an example 9 images out of a total of 850 that were compressed, we calculate the average of the compression rate, the maximum compression and the minimum compression. Examples of such images are presented in Figure 24. In this table, also, it is observed that the 2D CDT has the best compression.

Figure 25 and Figure 26 correspond to examples of knee X-ray images; of a total of 517 there are 2 subsets of 452 and 65 elements, for 1 and 2 knees respectively. They are presented in 24-bit PNG format. Table 8 presents some examples of the average compression, the maximum compression, and the minimum compression ratio for this image dataset. Also, in this medical image dataset, we can observe that the 2D TDC has the best lossless compression.

## 5. Discussion

The experiments and the results carried out and shown in this paper reveal that the 2D TDC has outstanding advantages over commercial algorithms such as JPEG, TIFF, and PNG, which makes it a very advantageous option for use in medical imaging and also for other types of images. Despite the fact that it was not the best lossless compression algorithm for grayscale images, it should not necessarily be discarded for use in these images. As mentioned above, 8-bit grayscale images by nature do not have many bits to define the gray tone; that is, they are light images in terms of bit size. For this reason, the 2D TDC can be utilized to compress grayscale images. The way in which the 2D TDC is designed guarantees that the more information there is, the greater the compression will be, asl confirmed by the tables and figures that refer to 16- and 24-bit images. We believe that 2D TDC is an excellent lossless compression option for 16-bit and 24-bit images.

When performing a transformation on the pixels in a digital image, it can produce scattered values between a known range, even though the total of possible values is unknown, which is a problem. The 2D TDC manages to identify the values that are obtained. This is an advantage offered by the 2D TDC. For example, for the 8-bit representation, the result of the 2D TDC corresponds only to $2^{N+1}+1$ possible values within an interval of $\left[-2^{N},2^{N}\right]$ where *N* is the number of bits used to represent each pixel.

In Figure 13 and Figure 14, the TDC can be visually observed as an image in which the grayscale areas are very similar, and the greatest compression is generated. On the other hand, the compression is more minute where there are gradients or contours. In Figure 14, it can be seen that 2D TDC can be applied to the compressed image and further compression is achieved. With the proposed algorithm, we achieve the decorrelation of the information in a simple and fast way compared to commercial algorithms.

The algorithms compared with the 2D TDC, are considered as standard formats, so they are fully developed and optimized. The above gives the TDC an edge because it is not optimized. For TDC, in this research, only Huffman coding was used initially to test its behavior and still obtained excellent results, beating PNG and TIFF. We propose, as a future work, to investigate and apply other coding methods to optimize the algorithm and significantly improve the results obtained in this paper. The latter allows us to propose the following hypothesis: Because the TDC generates values only within a specific interval and uses optimal coding methods, it is possible to create an efficient coding model that contains only the values obtained and not an undefined range of values as happens with other transformations.

There are recent algorithms that are efficient in lossless compression as is the case of Kabir [61]. They perform lossless compression through axis-based transformations and predictions along with entropy coding, achieving average compression ratios of 2.06. Another algorithm is the one registered in [62], where they proposed a joint compression and encryption scheme based on adaptive lossless image codec (CALIC) and hyperchaos, indicating that it achieves compression ratios of up to 15.87. On the other hand, in [63] they use the Golomb–Rice prediction and coding techniques applied to a chip specifically designed to achieve compression obtaining an average compression ratio of 1.53. On our part, the highest lossless compression obtained was 7756 as can be seen in Table 3. When compared to the three algorithms mentioned above, it is clear that our algorithm does not compress the most, but does commpress the second most. This may be due to other situations; for example, [61] makes the tests with a set of pixelated images, whereas we use datasets with large images—on average 4000 × 2000 pixels as shown in Figure 15 and Figure 16. It would be ideal to compare the Difference Transform with state-of-the-art algorithms under the same computing conditions and with the same image sets to determine which would be the best compressor.

One of the contributions we present in this study is that we propose an algorithm that differs from the existing ones because we show a transform that is simple to implement, fast in its execution, and capable of compressing more than the standard compression algorithms (which are the algorithms against which all new compression algorithms are compared). Another aspect that differentiates our algorithm from others is that it can replace the Wavelets Transform and the Cosine Transform, which are widely used for new compression methods such as those mentioned in the related works section. That is the hallmark of this paper—to present a new, lossless compression algorithm that can be useful in areas such as medicine where data sensitivity matters.

## 6. Conclusions

In this paper, a new transformation algorithm for digital images was presented: the Difference Transform for 1 and 2 dimensions applied to lossless compression, where medical image datasets were used. Moreover, non-medical images were also used to demonstrate that the TDC is competitive with image algorithms such as the JPEG-LS. Medical datasets are in PNG and TIFF formats. Through the results shown, the TDC proved to have higher lossless compression than the commercial algorithms (JPEG-LS, TIFF, and PNG). The results confirm that 2D TDC is recommended for use in medical images, where the images contain a lot of information and need to occupy as little space as possible for their processing, display, and transmission.

Further research for the application of the Difference Transform concerns the content of 360° images. This issue is novel because it allows the conversion of 360° images into metric products [64]. These 360° images have ultra-high resolution that maps to the two-dimensional plane and conforms to existing encoding standards for higher transmission speed. For example, [65] shows an evaluation framework for the coding performance of various projection formats including graphic formats such as JPEG and JPEG2000, presenting quality 2D metrics for measuring distortion in 360° images. Reference [66] presents an international JPEG 360° development that proposes a compatibility standardization between cameras and software. For this proposal, Huffman coding and the Cosine Transform are used. In [67], an effective algorithm is proposed to evaluate 360° omnidirectional image quality without reference using multifrequency and local information. They decompose the projected equirectangular projection maps into Wavelet subbands; with the proposed multifrequency information measurement and the local–global naturalness measurement, a support vector regression is used as the final image quality regressor. Because the Difference Transform outperforms formats like JPEG2000 and can replace the Cosine Transform and the Wavelet Transform, it would be an excellent opportunity to apply it to 360° images under the same metrics to determine the quality of the image.

Another investigation would involve image quality. There is one study by Wei Zhou et al. which proposes a method to evaluate image quality through super-resolution algorithms. This involves examining a unique image in a two-dimensional space of structural fidelity versus statistical naturalness [68]. Moreover, to improve perception and image quality, Xin Deng and et al. propose a method based on Wavelet domain-style transfer, which manages to improve the compensation of perception distortion. They propose the use of the 2D stationary Wavelet Transform to decompose an image into low- and high-frequency subbands, achieving interesting results [69]. This gives us the opportunity to apply the Difference Transform to 360° image standards and test the effectiveness of the image quality.

## Figures and Tables

**Figure 1 entropy-24-00951-f001:**
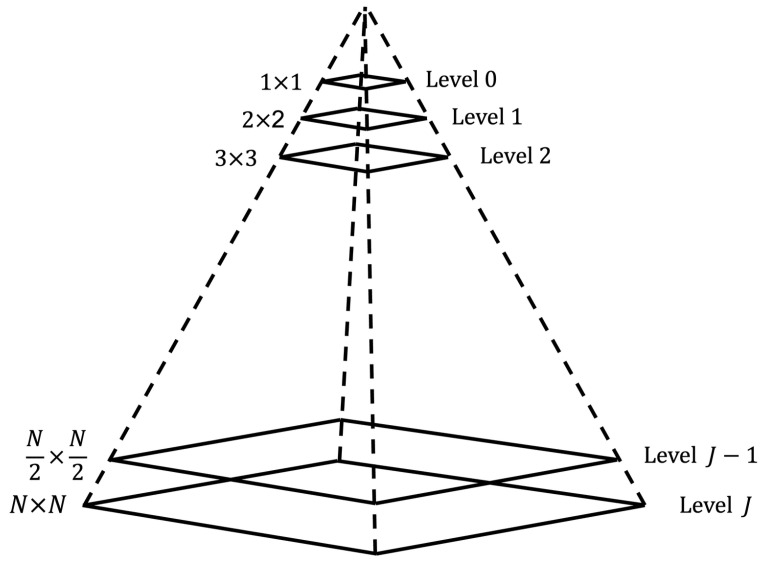
An image pyramid structure.

**Figure 2 entropy-24-00951-f002:**
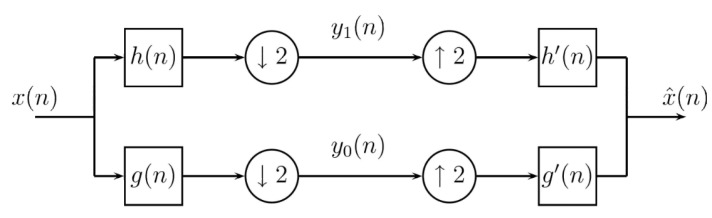
Subband coding scheme.

**Figure 3 entropy-24-00951-f003:**
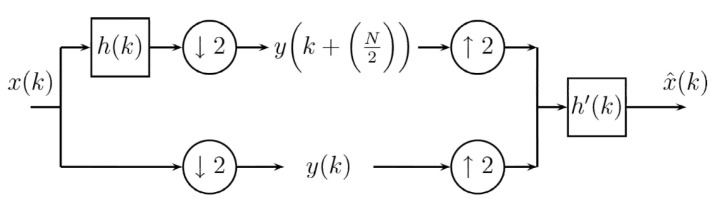
Coding scheme for Difference Transform.

**Figure 4 entropy-24-00951-f004:**
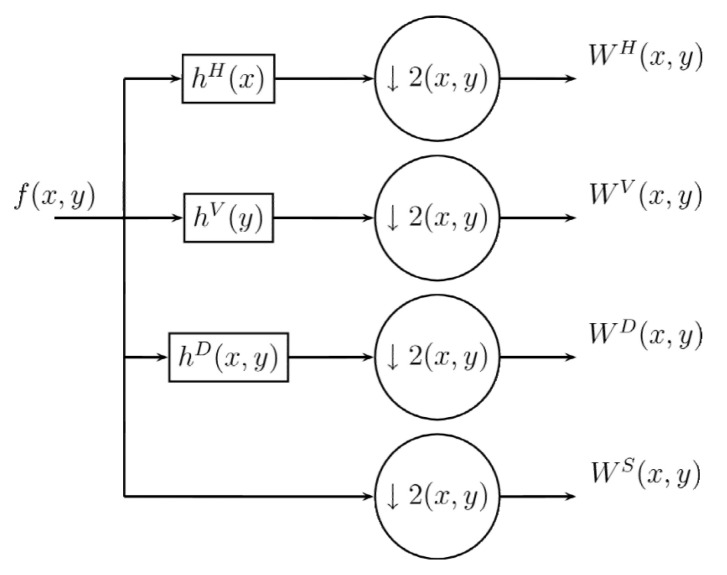
Coding procedure transform 2D difference.

**Figure 5 entropy-24-00951-f005:**
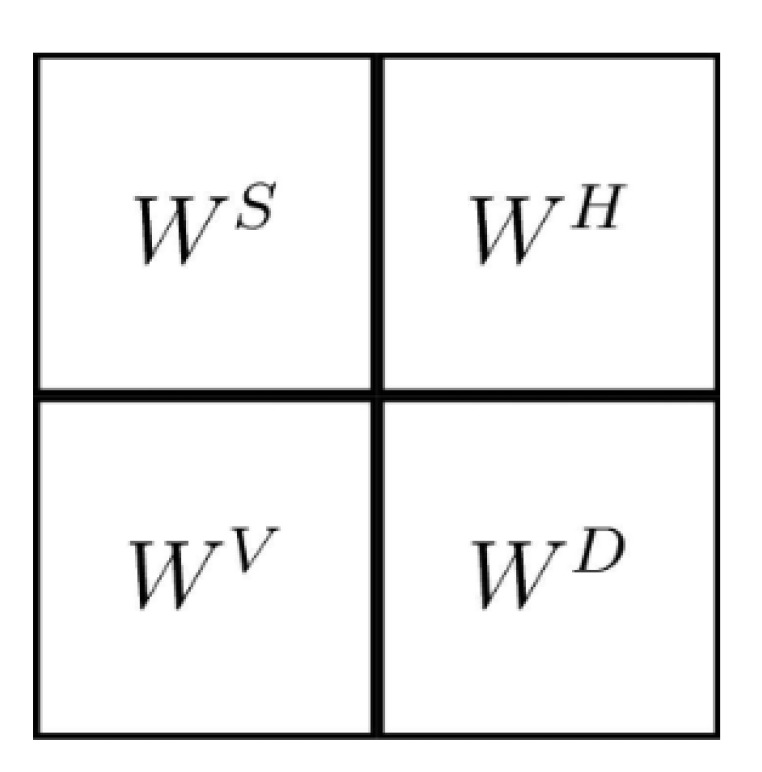
Arrangement of subimages obtained for 2D Difference Transform.

**Figure 6 entropy-24-00951-f006:**
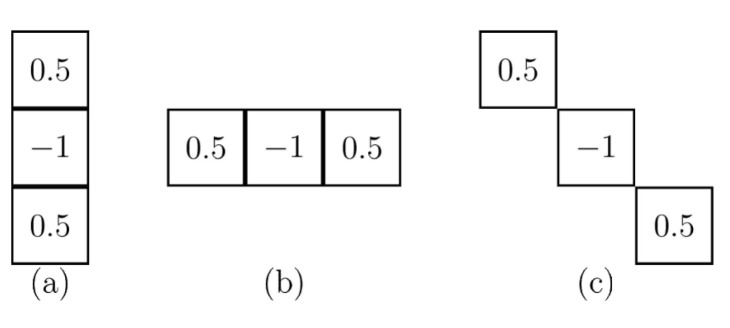
The filter structure (**a**) $h^{H}$, (**b**) $h^{V}$, (**c**) $h^{D}$.

**Figure 7 entropy-24-00951-f007:**
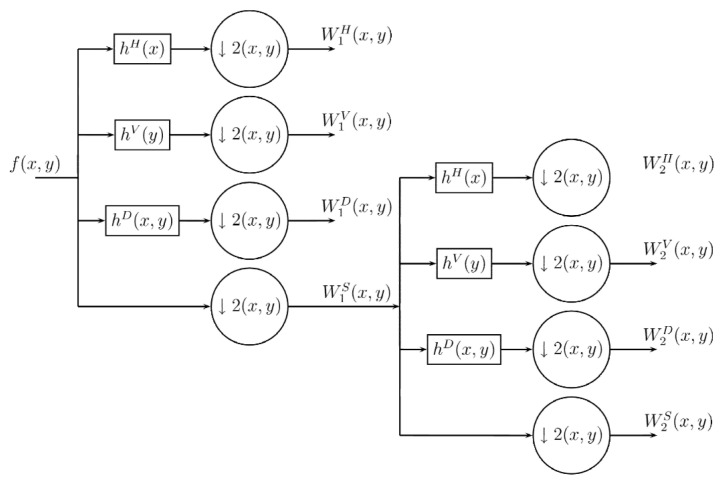
Multiresolution decomposition procedure for 2D Difference Transform.

**Figure 8 entropy-24-00951-f008:**
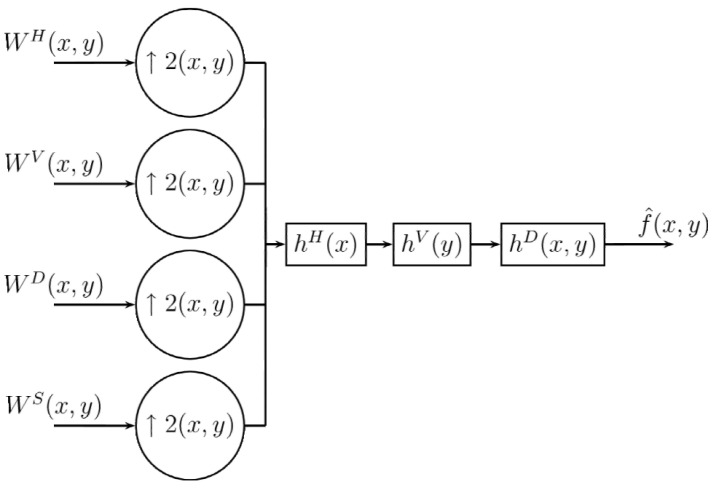
Decoding procedure for 2D Difference Transform.

**Figure 9 entropy-24-00951-f009:**
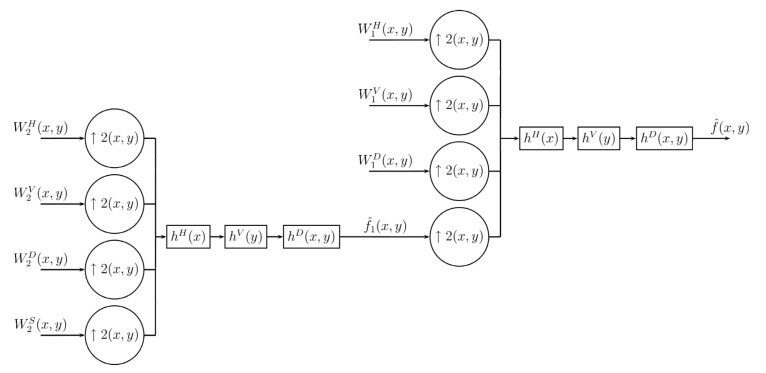
Multiresolution decoding procedure for 2D Difference Transform.

**Figure 10 entropy-24-00951-f010:**
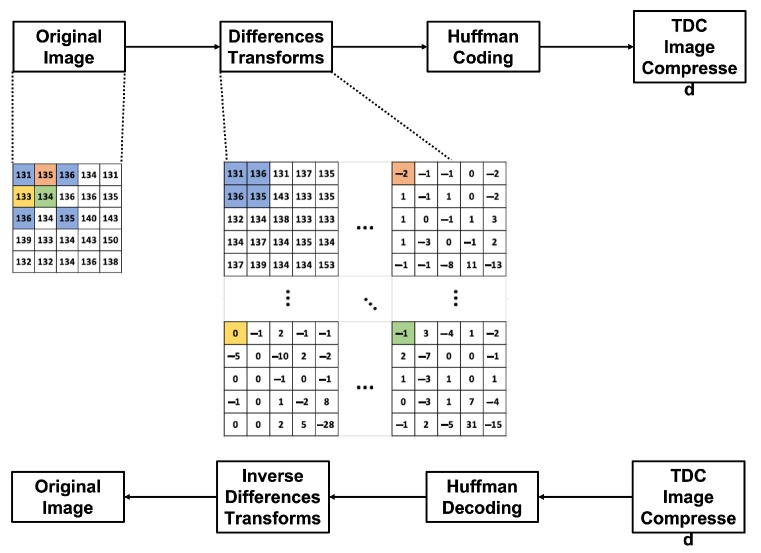
Difference Transform compression/decompression model, a numerical example.

**Figure 11 entropy-24-00951-f011:**
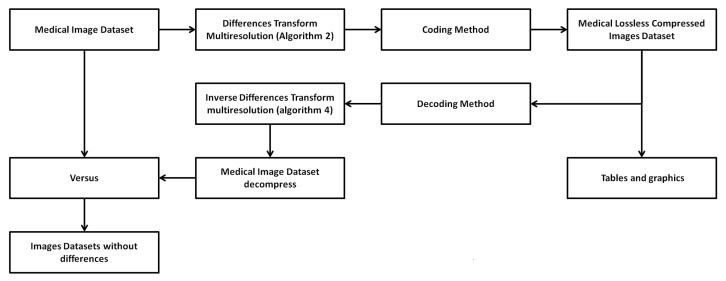
Process to generate the medical image dataset with lossless compression.

**Figure 12 entropy-24-00951-f012:**
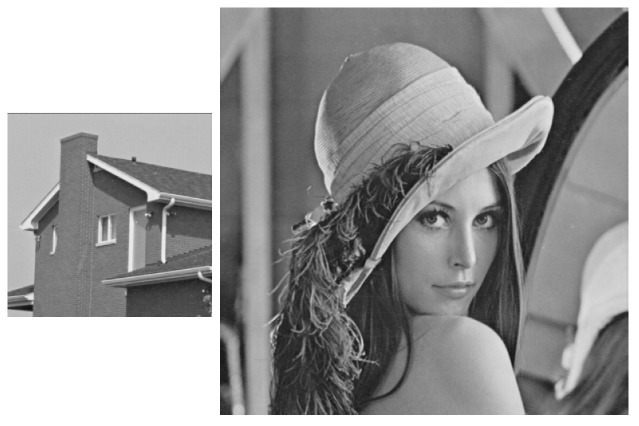
Original images.

**Figure 13 entropy-24-00951-f013:**
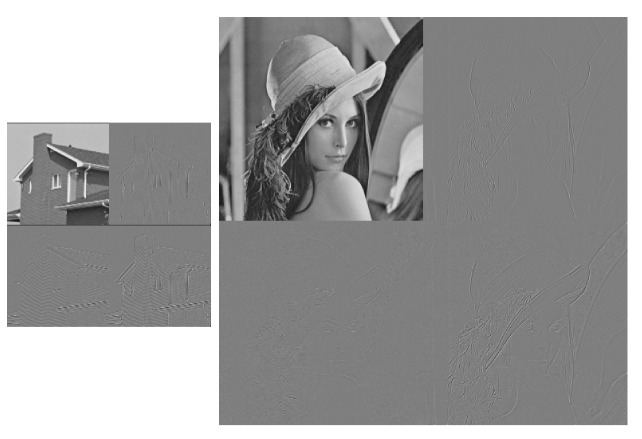
Result of applying the 2D TDC for the first time.

**Figure 14 entropy-24-00951-f014:**
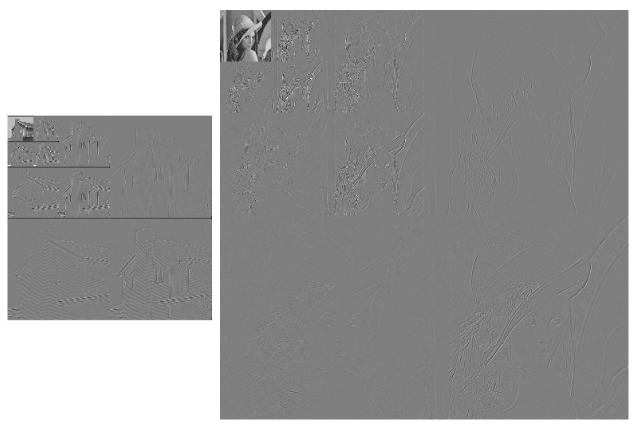
Result of applying the 2D TDC three times.

**Figure 15 entropy-24-00951-f015:**
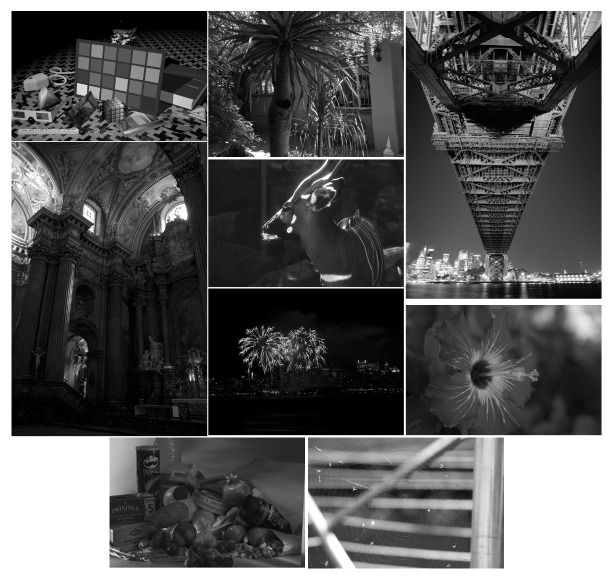
Set of 8-bit and 16-bit grayscale images.

**Figure 16 entropy-24-00951-f016:**
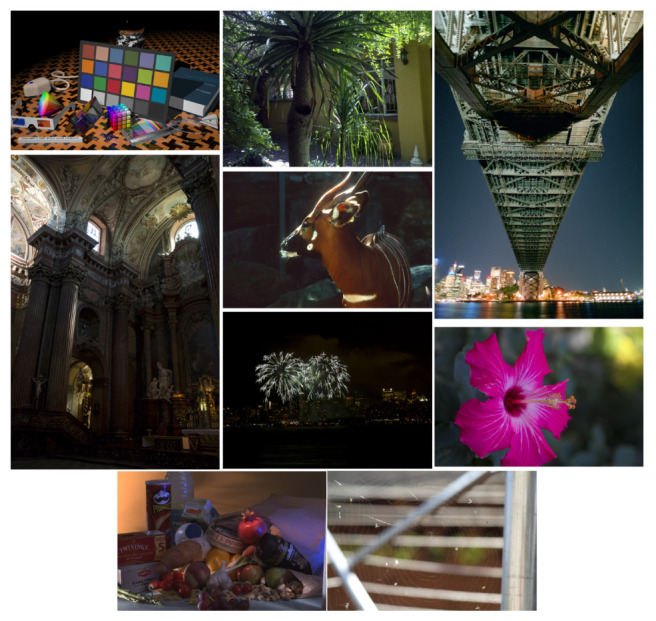
Set of 8-bit and 16-bit color images.

**Figure 17 entropy-24-00951-f017:**
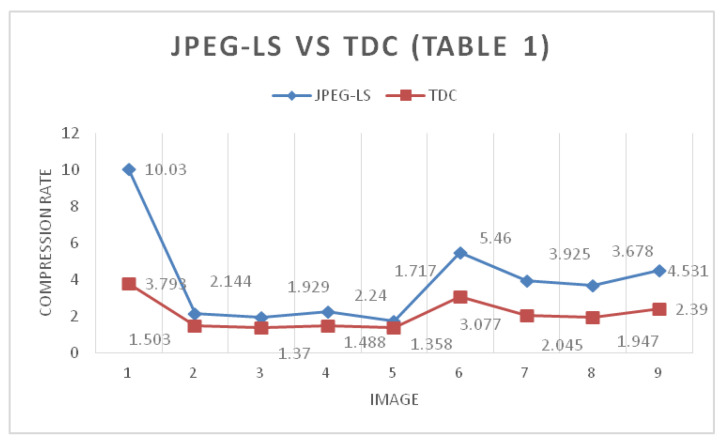
Compression rate difference between JPEG-LS and TDC with 8-bit grayscale images.

**Figure 18 entropy-24-00951-f018:**
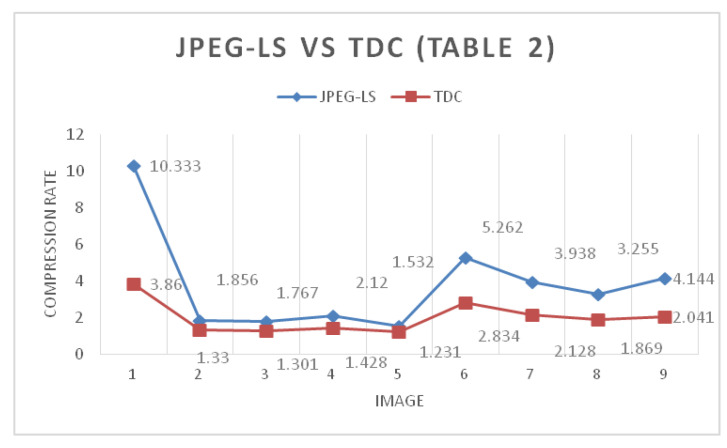
Compression rate difference between JPEG-LS and TDC with 8-bit color images.

**Figure 19 entropy-24-00951-f019:**
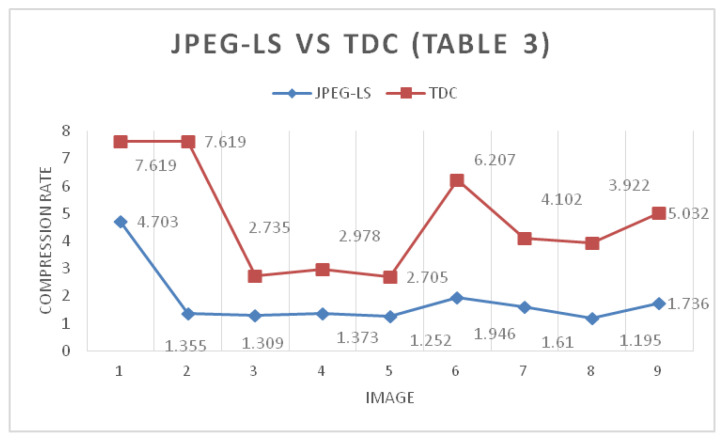
Compression rate difference between JPEG-LS and TDC with 16-bit gray-scale images.

**Figure 20 entropy-24-00951-f020:**
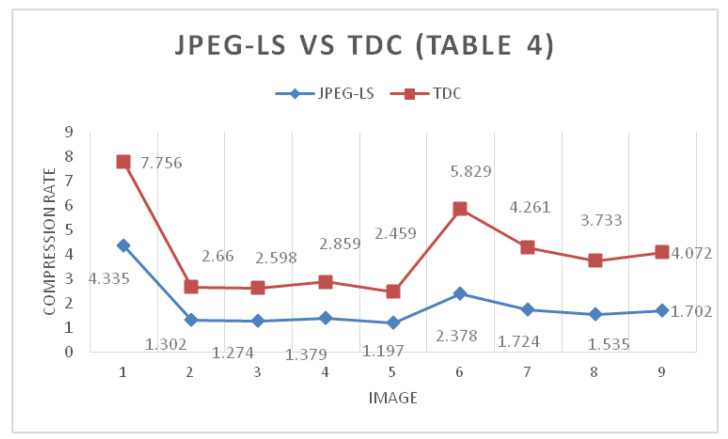
Comparison JPEG-LS VS TDC (Table 4).

**Figure 21 entropy-24-00951-f021:**
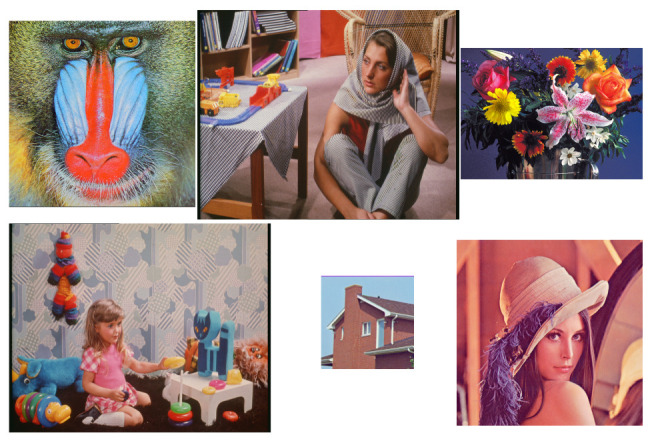
Set of 24-bit color images.

**Figure 22 entropy-24-00951-f022:**
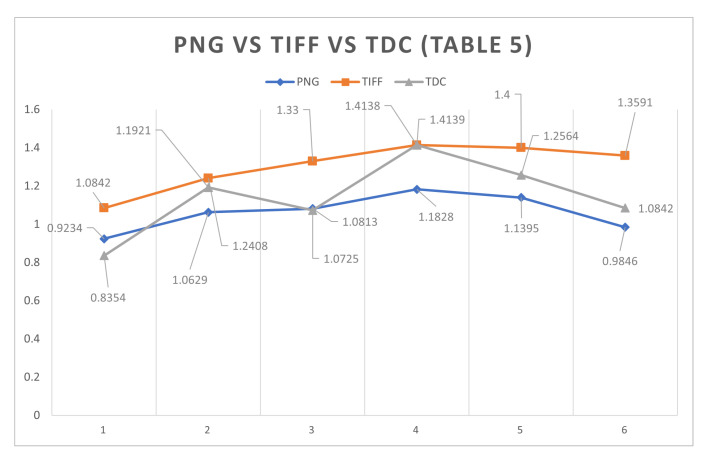
Comparison between JPEG-LS and 2D TDC (Table 5).

**Figure 23 entropy-24-00951-f023:**
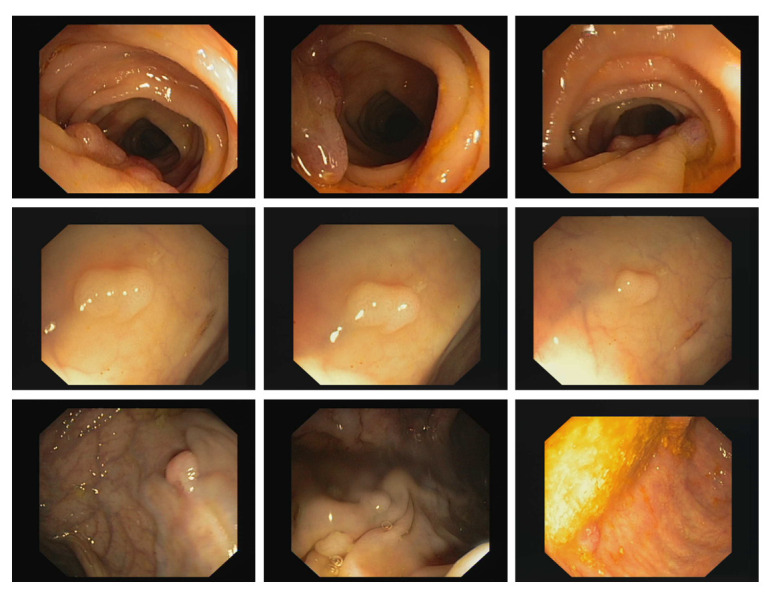
Set of 24-bit color images.

**Figure 24 entropy-24-00951-f024:**
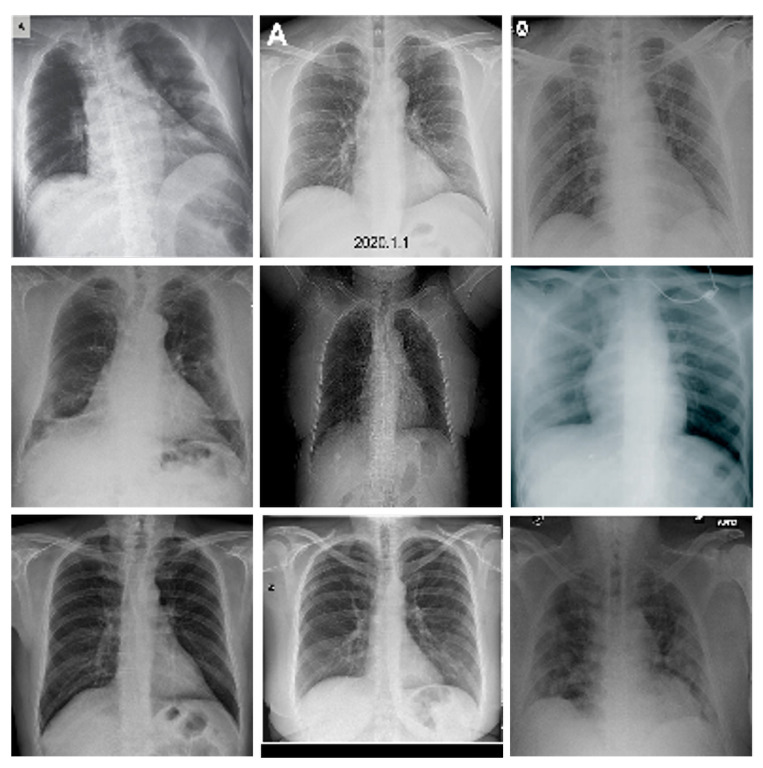
COVID-19 Chest images dataset X-ray (examples).

**Figure 25 entropy-24-00951-f025:**
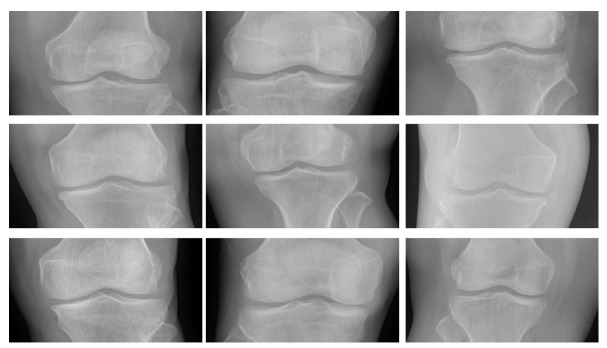
Knee X-ray dataset (1 knee).

**Figure 26 entropy-24-00951-f026:**
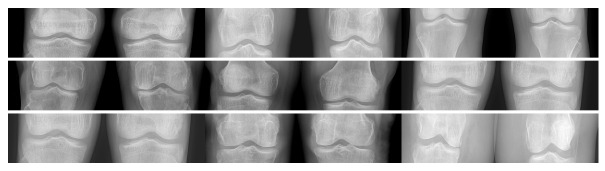
Knee X-ray dataset (2 knee).

**Table 1 entropy-24-00951-t001:** Compression rate in 8-bit grayscale. (common formats).

Image Name (Dimensions)	JPEG-LS	JPEG 2000	Lossless JPEG	PNG	TDC
1. artificial.pgm (2048 × 3072)	10.03	6.72	4.884	8.678	3.793
2. big_tree.pgm (4550 × 6088)	2.144	2.106	1.806	1.973	1.503
3. bridge.pgm (4049 × 2749)	1.929	1.91	1.644	1.811	1.370
4. cathedral.pgm (4049 × 2749)	2.241	2.16	1.813	2.015	1.488
5. deer.pgm (2641 × 4043)	1.717	1.748	1.583	1.713	1.358
6. fireworks.pgm (2352 × 3136)	5.46	4.853	3.355	4.095	3.077
7. flowers_foveon.pgm (1512 × 2268)	3.925	3.65	2.97	3.054	2.045
8. hdr.pgm (2048 × 3072)	3.678	3.421	2.795	2.857	1.947
9. spider_web.pgm (2848 × 4256)	4.531	4.202	3.145	3.366	2.390

**Table 2 entropy-24-00951-t002:** Compression rate in 8-bit RGB (common formats).

Image Name (Dimensions)	JPEG-LS	JPEG 2000	Lossless JPEG	PNG	TDC
1. artificial.pgm (2048 × 3072)	10.333	8.183	4.924	10.866	3.860
2. big_tree.pgm (4550 × 6088)	1.856	1.823	1.585	1.721	1.330
3. bridge.pgm (4049 × 2749)	1.767	1.765	1.553	1.686	1.301
4. cathedral.pgm (4049 × 2749)	2.12	2.135	1.734	1.922	1.428
5. deer.pgm (2641 × 4043)	1.532	1.504	1.407	1.507	1.231
6. fireworks.pgm (2352 × 3136)	5.262	4.496	3.279	3.762	2.834
7. flowers_foveon.pgm (1512 × 2268)	3.938	3.746	2.806	3.149	2.128
8. hdr.pgm (2048 × 3072)	3.255	3.161	2.561	2.653	1.869
9. spider_web.pgm (2848 × 4256)	4.411	4.209	3.029	3.365	2.041

**Table 3 entropy-24-00951-t003:** Compression rate in 16-bit Grayscale (common formats).

Image Name (Dimensions)	JPEG-LS	JPEG 2000	Lossless JPEG	PNG	TDC
1. artificial.pgm (2048 × 3072)	4.703	4.007	2.791	4.381	7.619
2. big_tree.pgm (4550 × 6088)	1.355	1.325	1.287	1.181	3.561
3. bridge.pgm (4049 × 2749)	1.309	1.279	1.244	1.147	2.735
4. cathedral.pgm (4049 × 2749)	1.373	1.337	1.293	1.191	2.978
5. deer.pgm (2641 × 4043)	1.252	1.241	1.225	1.132	2.705
6. fireworks.pgm (2352 × 3136)	1.946	1.809	1.740	1.604	6.207
7. flowers_foveon.pgm (1512 × 2268)	1.610	1.591	1.523	1.316	4.102
8. hdr.pgm (2048 × 3072)	1.195	1.563	1.491	1.297	3.922
9. spider_web.pgm (2848 × 4256)	1.736	1.771	1.554	1.367	5.032

**Table 4 entropy-24-00951-t004:** Compression rate in 16-bit RGB (common formats).

Image Name (Dimensions)	JPEG-LS	JPEG 2000	Lossless JPEG	PNG	TDC
1. artificial.pgm (2048 × 3072)	4.335	4.734	2.695	4.896	7.756
2. big_tree.pgm (4550 × 6088)	1.302	1.261	1.249	1,159	2.660
3. bridge.pgm (4049 × 2749)	1.274	0.1243	1.240	1.132	2.598
4. cathedral.pgm (4049 × 2749)	1.379	1.333	1.267	1.218	2.859
5. deer.pgm (2641 × 4043)	1.197	1.173	1.236	1.100	2.459
6. fireworks.pgm (2352 × 3136)	2.378	1.832	1.688	2.053	5.829
7. flowers_foveon.pgm (1512 × 2268)	1.724	1.664	1.494	1.371	4.261
8. hdr.pgm (2048 × 3072)	1.535	1.518	1.473	1.275	3.733
9. spider_web.pgm (2848 × 4256)	1.702	1.784	1.531	1.360	4.072

**Table 5 entropy-24-00951-t005:** Compression rate in 24-bit RGB (common formats).

Image Name (Dimensions)	PNG	TIFF	TDC
1. Baboon.bmp (512 × 512)	0.9234	1.0842	0.8354
2. Barbara.bmp (720 × 576)	1.0629	1.2408	1.1921
3. Flowers.bmp (500 × 362)	1.0813	1.3300	1.0725
4. Girl.bmp (720 × 576)	1.1828	1.4138	1.4139
5. House.bmp (256 × 256)	1.1395	1.4000	1.2564
6. Lenna.bmp (512 × 512)	0.9846	1.3591	1.0842

**Table 6 entropy-24-00951-t006:** Compression rate medical images CVC-ClinicDB.

Image Name (Dimensions)	PNG	TDC
1. 1.tif (384 × 288)	3.4766	5.5759
2. 10.tif (384 × 288)	3.6817	5.6551
3. 20.tif (384 × 288)	4.3624	6.0639
4. 30.tif (384 × 288)	3.7004	5.6567
5. 40.tif (384 × 288)	4.1710	6.5407
6. 50.tif (384 × 288)	5.1542	7.2669
7. 60.tif (384 × 288)	6.2902	7.1897
8. Average (384 × 288)	3.9926	5.8980
9. Maximum (384 × 288)	2.7035	4.7716
10. Minimum (384 × 288)	6.8059	7.9004

**Table 7 entropy-24-00951-t007:** Compression rate medical images Covid-19 Chest X-Ray.

Image Name (Dimensions)	TIFF	TDC
1. Normal.png (128 × 128)	0.4447	1.8100
2. Normal_64.png (128 × 128)	0.4094	1.5850
3. Normal_115.png (128 × 128)	0.3385	1.5111
4. Normal_199.png (128 × 128)	0.3684	1.5521
5. Normal_255.png (128 × 128)	0.3726	1.6125
6. Normal_459.png (128 × 128)	0.3875	1.4296
7. Normal_629.png (128 × 128)	0.3877	1.4990
8. Average (128 × 128)	0.3790	1.5994
9. Maximum (128 × 128)	0.8928	3.4762
10. Minimum (128 × 128)	0.2619	1.3074

**Table 8 entropy-24-00951-t008:** Compression rate medical images knee X-ray dataset (1 knee).

Image Name (Dimensions)	TIFF	TDC
1. Normal G0(4).png (300 × 162)	0.2113	1.2746
2. Normal G0(83).png (300 × 162)	0.1993	1.2493
3. Normal G0(120).png (300 × 162)	0.4523	3.3081
4. Normal G0(217).png (300 × 162)	0.1677	1.1702
5. Normal G0(270).png (300 × 162)	0.1974	1.1975
6. Normal G0(336).png (300 × 162)	0.1438	1.0578
7. Normal G0(441).png (300 × 162)	0.4155	1.1828
8. Average (300 × 162)	0.2471	1.3761
9. Maximum (300 × 162)	1.1087	3.2872
10. Minimum (300 × 162)	0.1283	0.9140

**Table 9 entropy-24-00951-t009:** Compression rate medical images knee X-ray dataset (2 knees).

Image Name (Dimensions)	TIFF	TDC
1. Normal G0(452).png (640 × 161)	0.1829	1.1113
2. Normal G0(452).png (640 × 161)	0.1126	1.2929
3. Normal G0(452).png (640 × 161)	0.1053	1.3063
4. Normal G0(452).png (640 × 161)	0.0986	1.2976
5. Normal G0(452).png (640 × 161)	0.0969	1.2983
6. Normal G0(452).png (640 × 161)	0.0948	1.2635
8. Average (640 × 161)	0.1213	1.3124
9. Maximum (640 × 161)	0.5260	3.2615
10. Minimum (640 × 161)	0.0921	1.0967

## Data Availability

The images in Figure 15 and Figure 16 were taken from: https://imagecompression.info/test-images/ (accessed on 26 May 2022). The images in Figure 23 were taken from: https://www.kaggle.com/datasets/balraj98/cvcclinicdb (accessed on 26 May 2022). The images in Figure 24 were taken from: https://www.kaggle.com/datasets/sachinkumar413/cxr-2-classes (accessed on 26 May 2022). The images in Figure 25 and Figure 26 were taken from: https://www.kaggle.com/datasets/tommyngx/digital-knee-xray (accessed on 26 May 2022).

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
