# Peer review of "Lossless Medical Image Compression by Using Difference Transform"

_entropy, 2022, doi:10.3390/e24070951_

Round 1

Reviewer 1 Report

This paper proposes a lossless medical image compression method based on difference transform. Experimental results validate the effectiveness of the proposed method. Detailed comments are listed as follows:

1. In the related work, it is suggested that the existing works can be classified and introduced from natural image compression to medical image compression, instead of listing them one by one. Moreover, after stating the existing works, the drawbacks should be pointed out to introduce the proposed model.

2. The paper focuses on medical images, but natural images are also used to validate the results. Please clarify this point. And why the proposed method is more appropriate for dealing with medical images.

3. A figure about the proposed framework would be helpful to understand the key idea.

4. The main contributions should be listed in the end of the introduction part.

5. In the beginning of the results, each dataset should be described. Ablation study would be helpful to validate the performance of difference transform.

6. The transform used in other image processing fields are suggested to be reviewed, including No-reference quality assessment for 360-degree images by analysis of multifrequency information and local-global naturalness, Wavelet domain style transfer for an effective perception-distortion tradeoff in single image super-resolution, Image super-resolution quality assessment: Structural fidelity versus statistical naturalness, etc.

7. Please further improve the language and presentation, such as difference or differences, transform or transforms. And the image transform only appears in the keyworks. The medical image dataset is recommended to remove from the keywords. The tables are too big.

Author Response

We attach a file attending to the observations

Reviewer 2 Report

In this paper, the authors have presented a lossless digital image compression method using the Difference Transform.The following issues the authors should revise.

1. The abstract can be improved considerably. It should be divided into three parts, the first should introduce your work in 2 to3 lines. Next, 2 to 3 lines should discuss your method/algorithm. Finally, the results and comparative summary must be highlighted with the considered measures.

2. Further, the research issue and objective is not clear.

3. The literature review section needs to discuss more in detail issues of existing works.

4. The proposed work is a good and interesting one.

5. A numerical illustration is required to understand the proposed work completely.

6. Some more measures or parameters need to be considered for evaluation.

7.  Comparision should be done with recent works.

8. Discuss the future work and implications of the study.

Author Response

(The authors gave the same response as above.)

Round 2

Reviewer 1 Report

The authors addressed some of my comments.

1. But after reading the contributions and the proposed framework, one main concern is how the proposed model differs from the existing ones? Even there lack comparisons with state-of-the-art.

2. The suggested literature as shown below also uses the transform, at least the authors should mention them in the texts for future work or existing work.

No-reference quality assessment for 360-degree images by analysis of multifrequency information and local-global naturalness, Wavelet domain style transfer for an effective perception-distortion tradeoff in single image super-resolution, Image super-resolution quality assessment: Structural fidelity versus statistical naturalness, etc.

Author Response

We attach a file with the observations attended

Reviewer 2 Report

I am satisfied with the response

Author Response

Thank you very much for your contributions

Round 3

Reviewer 1 Report

Thanks for your revisions. The paper can be accepted, and the figures are suggested to improve the clarity since the current version is quite blurry.